# Synthesis, Structural and Pharmacological Characterizations of CIC, a Novel α-Conotoxin with an Extended N-Terminal Tail

**DOI:** 10.3390/md19030141

**Published:** 2021-03-02

**Authors:** Julien Giribaldi, Yves Haufe, Edward R. J. Evans, David T. Wilson, Norelle L. Daly, Christine Enjalbal, Annette Nicke, Sébastien Dutertre

**Affiliations:** 1Institut des Biomolécules Max Mousseron, UMR 5247, Université de Montpellier, CNRS, ENSCM, 34095 Montpellier, France; julien.giribaldi@umontpellier.fr (J.G.); christine.enjalbal@umontpellier.fr (C.E.); 2Walther Straub Institute of Pharmacology and Toxicology, Faculty of Medicine, LMU Munich, Nußbaumstraße 26, 80336 Munich, Germany; yves.haufe@lrz.uni-muenchen.de (Y.H.); annette.nicke@lrz.uni-muenchen.de (A.N.); 3Centre for Molecular Therapeutics, Australian Institute of Tropical Health and Medicine, James Cook University, Cairns, QLD 4878, Australia; edwardrobertjonathan.evans@my.jcu.edu.au (E.R.J.E.); david.wilson4@jcu.edu.au (D.T.W.); norelle.daly@jcu.edu.au (N.L.D.)

**Keywords:** conotoxin, peptide synthesis, NMR structure, electrophysiology, nicotinic acetylcholine receptors

## Abstract

Cone snails are venomous marine predators that rely on fast-acting venom to subdue their prey and defend against aggressors. The conotoxins produced in the venom gland are small disulfide-rich peptides with high affinity and selectivity for their pharmacological targets. A dominant group comprises α-conotoxins, targeting nicotinic acetylcholine receptors. Here, we report on the synthesis, structure determination and biological activity of a novel α-conotoxin, CIC, found in the predatory venom of the piscivorous species *Conus catus* and its truncated mutant Δ-CIC. CIC is a 4/7 α-conotoxin with an unusual extended N-terminal tail. High-resolution NMR spectroscopy shows a major influence of the N-terminal tail on the apparent rigidity of the three-dimensional structure of CIC compared to the more flexible Δ-CIC. Surprisingly, this effect on the structure does not alter the biological activity, since both peptides selectively inhibit α3β2 and α6/α3β2β3 nAChRs with almost identical sub- to low micromolar inhibition constants. Our results suggest that the N-terminal part of α-conotoxins can accommodate chemical modifications without affecting their pharmacology.

## 1. Introduction

Conotoxins are short peptides (8 to 32 residues) highly constrained by disulfide bridges (1 to 4 disulfide bonds) found in predatory marine cone snail venoms [1]. Cone snails have evolved into more than 800 different species [2]; their fast-acting venom is composed of 100–200 different conotoxins and has been tailored for predation and defense purposes [3]. There is relatively low overlap between the toxins present within different cone snail species [4], and it is estimated that cone snail venoms may overall contain at least 70,000 mostly neuroactive peptides, of which less than 0.1% have currently been pharmacologically characterized [5]. Moreover, their small size, structural stability and target specificity are valuable assets that make them important pharmacological probes or even drug leads [6]. Based on their conserved signal sequence, conotoxins are grouped into various gene superfamilies and further classified according to their cysteine frameworks and their biological activity [5].

Members of the α-subfamily specifically target neuronal and muscle-type nicotinic acetylcholine receptors (nAChRs) [1]. In addition, a pattern is emerging based on the number of amino acids between the cysteine residues: 3/5 α-conotoxins are most likely to target muscle-type nAChRs, while 4/7 α-conotoxins target α7 and/or α3β2* neuronal nAChRs [7]. However, in most α-conotoxins, the number of residues outside the cysteine loops is usually limited to one or two amino acids (Figure 1A). Based on Conoserver data [8,9], α-conotoxin GID is the only described 4/7 α-conotoxin showing a significant N-terminal extension (>2 residues) and is a potent blocker of α3β2 and α7 (low nanomolar IC_50_) nAChRs, but also shows a significant and noteworthy inhibition of α4β2 nAChR subtypes (IC_50_ 152 nM), which appear to be “α-conotoxin-resistant” receptors [1,10]. Considering the important role of α4 and β2 subunits in addiction, pain and cognition, which has been revealed by knockout studies [11], α4β2 nAChR-selective ligands would be very valuable pharmacological tools. Interestingly, deletion of the N-terminal sequence of GID (∆-1-4) leads to loss of activity at the α4β2 subtype but has no effect on the IC_50_ value at α3β2 and α7 subtypes (but affects the dissociation rate at these receptors). These findings suggest that the N-terminal residues are important for both selectivity and affinity [10]. However, despite these effects, the N-terminal tail of GID appears to be disordered in solution. 

Here, we describe a new α-conotoxin, CIC, which was previously found in the proteo-transcriptomic study of *Conus catus* and identified as a component of the predation-evoked venom [12]. The native CIC has never been isolated from the venom and tested nor its sequence chemically synthesized. CIC shows high sequence similarity to PeIA and MII α-conotoxins, both potent antagonists of α3β2 nAChR subtypes with IC_50_ values in the low nanomolar range [13,14]. However, unlike every other characterized 4/7 α-conotoxins, CIC displays an extended N-terminal tail comprising six residues. Therefore, the aim of this study was to investigate the role of this unusual N-terminal part on the three-dimensional structure and on the pharmacological activity. To this end, native CIC conotoxin, as well as a deletion variant with no N-terminal tail (Ala1-Thr6 removed) named ∆-CIC, were synthesized and both pharmacologically and structurally characterized.

## 2. Results

### 2.1. Chemical Synthesis of CIC and ∆-CIC

CIC and ∆-CIC were synthesized manually using Fmoc-SPPS after anchoring of the first C-terminal residue on a chloro-(2′-chloro) trityl resin. Considering its sequence homology to other 4/7 α-conotoxins (Figure 1A), which all display the canonical globular disulfide bond connectivity CysI-CysIII, CysII-CysIV, linear peptides were folded using a regioselective folding strategy to obtain the globular isomer. Briefly, the first disulfide bridge (CysII-CysIV) is formed by oxidation of the free cysteine residues in an aqueous basic buffer assisted by DTP (2,2′-Dithiopyridine), and the second disulfide bond (CysI-CysIII) is formed by deprotection/oxidation of the Acm protecting groups [15]. The homogeneity of folded toxins was assessed by analytical RP-HPLC coupled to ESI–MS (Figure 1B). ESI–MS(+) confirmed a monoisotopic mass of 1602.90 g/mol (calculated 1602.59 g/mol) for ∆-CIC and 2106.27 g/mol (calculated 2106.29 g/mol) for CIC. Overall, 7.7 mg (yield 3.1% from resin loading) and 9.9 mg (yield 3.2% from resin loading) of pure (>95% based on LC/ESI–MS) ∆-CIC and CIC were obtained, respectively. 

### 2.2. Electrophysiology of CIC and ∆-CIC on Neuronal nAChRs

α-conotoxins PeIA [13] and MII [14], which are closely related (based on the primary structure, Figure 1A) to CIC, are potent inhibitors of α3β2 nAChRs subtypes, while the N-terminally extended GID conotoxin inhibits α3β2, α7 and α4β2 nAChR subtypes (Table 1). The latter is also inhibited to a limited extent by MII. Because of these similarities, the antagonistic potency of CIC and ∆-CIC was firstly investigated using two-electrode voltage-clamp analysis on α3β2, α7, and α4β2 rat nAChRs subtypes. Whereas no inhibition was detected at the α7 and α4β2 subtypes, CIC and Δ-CIC inhibited the rat α3β2 neuronal nAChR subtype with rather high IC_50_ values of 3.51 µM and 4.56 µM, respectively. At the closely related α6/α3β2β3 subtype, CIC and Δ-CIC showed IC_50_ values of 0.94 µM and 1.00 µM, respectively (Figure 2, Table 1). Considering that these values are approximatively 1000-fold higher than those of the similar (based on sequence) α-conotoxin MII, we assumed major structural differences and also tested CIC and Δ-CIC on additional rat nAChRs such as α4β4, α3β4, α2β2, α2β4 and (α1)2β1δγ subtypes as well as human α9α10 subtype. It is noteworthy that no inhibition was detected at concentrations up to 10 µM on any of the additional nAChRs tested. In summary, despite some similarity to potent conotoxins, both CIC and ∆-CIC showed weak or no inhibition at common mammalian nAChR combinations (with no difference in dissociation behavior), and the quasi-identical values of inhibition constant between CIC and Δ-CIC at α3β2-containing nAChRs suggest that there is no major influence of the N-terminal tail on their affinity. 

**Table 1 marinedrugs-19-00141-t001:** IC_50_ values of the conotoxins discussed in the text on the different nAChR subtypes. N.D means not determined.

nAChR Subtypes	Toxin (IC_50_)			
	CIC	Δ-CIC	MII	GID
**α3β2**	3.51 µM	4.56 µM	3.7 nM	3.1 nM
**α6/α3β2β3**	1.03 µM	1.08 µM	390 pM	N.D
**α7**	>10 µM	>10 µM	200 nM (56% inhibition)	4.5 nM
**α4β2**	>10 µM	>10 µM	200 nM (30% inhibition)	152 nM
**α4β4**	>10 µM	>10 µM	200 nM (4% inhibition)	>5 µM
**α3β4**	>10 µM	>10 µM	200 nM (15% inhibition)	>5 µM
**α2β2**	>10 µM	>10 µM	200 nM (20% inhibition)	N.D
**α2β4**	>10 µM	>10 µM	200 nM (4% inhibition)	N.D
**(α1)2β1δγ**	>10 µM	>10 µM	200 nM (11% inhibition)	N.D
**α9α10**	>10 µM	>10 µM	>1 µM	N.D
**References**	This work	This work	[14,16,17,18]	[10]

### 2.3. Three-Dimensional Structure Determination of CIC and ∆-CIC

High-resolution NMR spectroscopy allowed the determination of the three-dimensional structures of both CIC and Δ-CIC (Figure 3). The structure statistics are given in Table 2. The calculated structures of CIC are well-defined with an RMSD value over backbone atoms, excluding the N-terminal tail, of 0.29 ± 0.16 Å (residues 7–21). CIC exhibits two 3^10^ helices from residues Thr6 to Ser9 and Pro11 to Gln16. Interestingly the N-terminal tail displays a relatively well-defined structure. However, Δ-CIC lacks the first helix and has only one 3^10^ helix from residues Pro5 to Val9, suggesting that the N-terminal tail plays a role in inducing the first helix. Counterintuitively, the calculated structures of Δ-CIC are less defined and display a higher RMSD value (2.06 ± 0.73 Å, over backbone atoms) than CIC, which indicates a stabilizing role of the N-terminal tail. Moreover, the lack of the N-terminal tail in Δ-CIC appears to change the overall structure, with the C-terminal region (Ser18-Cys21) in a different orientation with respect to the helix.

## 3. Discussion

α-conotoxin CIC has been previously found in a proteo-transcriptomic study of the predation-evoked venom of *Conus catus* [12]. Since this toxin exhibited a typical α-conotoxin cysteine pattern -CC-C-C [5] along with an unusual N-terminal extended tail, we decided to investigate the influence of the N-terminal tail on the three-dimensional structure and on the pharmacological properties of the toxin. To this end, we synthesized the globular form (CysI-CysIII, CysII-CysIV) of the α-conotoxin CIC as well as a mutant variant named ∆-CIC in which the first six residues (Ala1-Thr6) were removed. Considering that the other known bioactive 4/7 α-conotoxins [5] mainly display the “native” globular fold, a regioselective folding strategy was applied. 

So far, α-GID [10] is the only described 4/7 α-conotoxin with an extended N-terminal tail. α-GID was isolated using assay-directed fractionation of *Conus geographus* crude venom and is a potent blocker of α3β2, α7 (low nanomolar IC_50_, Table 1) and α4β2 nAChR subtypes (IC_50_ 152 nM, Table 1). It is noteworthy that the deletion of the N-terminal sequence of GID (∆-1-4) leads to a significant loss of potency at the α4β2 subtype [10]. However, unlike α-GID, α-CIC showed no inhibition of α4β2 nAChR subtypes at concentrations up to 10 µM. Moreover, the structure of the N-terminal tail of CIC was found to be well-defined, which is in contrast to the disordered N-terminal tail of conotoxin GID [10] (Figure 4A). 

In addition to its N-terminal tail, α-CIC displays high sequence similarity to α-conotoxin MII, including the conservation of some critical residues (Figure 1A) that have been described to be responsible for its low nanomolar activity at α3β2* and α7 nAChRs [14,16,19]. Everhart et al. [18] performed an Ala-scan of α-conotoxin MII and showed that Asn5, Pro6 and His12 play a major role in α-conotoxin binding, probably via Asn5 and His12 electrostatic interactions with β2E61 and β2D169, as suggested by molecular modeling. Despite displaying all the sequence features required for the inhibitory potency and high sequence similarity with α-conotoxin MII, CIC and ∆-CIC were found to be ~1000-fold less potent at α3β2 and α6α3β2β3 nAChRs than MII (Table 1). This potency drop most likely arises from slight structural differences between MII and CIC/∆-CIC that weaken and/or prevent the critical interactions between the toxin and the nAChRs. To further assess the potency and selectivity profile of CIC and ∆-CIC, we have tested them at other common nAChR subunit combinations such as rat α4β4, α3β4, α2β2, α2β4 and (α1)2β1δγ subtypes as well as the human α9α10 subtype. Interestingly, no inhibition was detected at any of these receptors at concentrations up to 10 µM, highlighting the high selectivity profile of CIC and ∆-CIC for α3β2 and α6/α3β2β3 nAChRs. Moreover, our result suggests that there is no influence of the N-terminal tail on the activity at α3β2 and α6/α3β2β3 nAChRs since CIC and Δ-CIC display quasi-similar inhibition constants (Table 1, Figure 2). Thus, we suggest that the N-terminal tail could serve as a chemical platform for bioconjugation and that CIC bioconjugates could be developed into valuable pharmacological tools to probe α3β2* nAChRs.

While the N-terminal tail has no influence on CIC potency, it clearly plays two major roles at the structural level: (i) decrease of flexibility and (ii) stabilization of secondary structure. Indeed, Figure 3 clearly shows that the NMR structure of Δ-CIC is less defined than that of CIC, as highlighted by the RMSD values (Table 2). Moreover, the N-terminal tail is associated with an additional 3^10^ helix from residues Thr6 to Ser9, which contains the first two cysteine residues. This result suggests that the N-terminal tail of CIC plays a major role in the stabilization of the global structure. Additionally, the superposition of CIC and Δ-CIC reveals a major structural deviation of Δ-CIC (compared to CIC) in the C-terminal region (Figure 4B). Whereas the calculated RMSD value is 0.88 Å from residues Cys7 to Gln14 (CIC numbering), this value greatly increases to 6.29 Å from residues Cys7 to Cys21, i.e., when the C-terminal region is included. Interestingly, the C-terminal region is not only less structured but is in a different orientation compared to CIC, which could be attributed to the absence of the additional first helix (residues Thr6-Ser9), forcing the CysII-CysIV bridge into a different position compared to the CIC conotoxin. 

Overall, the role of the N-terminal tail in α-conotoxins remains poorly understood. During the writing of this article, a new conotoxin, Pl168, was identified in the transcriptome of *Conus planorbis*, which shows an unusual 4/8 loop framework. Pl168 also displays an extended N-terminal tail comprising five amino acids but did not show inhibitory potency on a range of nAChRs, Ca^2+^ and Na^+^ channels [20]. It cannot be excluded that these N-terminally extended conotoxins (Pl168 and CIC) target a different class of receptors or that they are designed to be effective on cone snail prey receptors, which likely display structural differences to mammalian receptors.

## 4. Materials and Methods

### 4.1. Abbreviations

ACh, Acetylcholine; Acm, acetamidomethyl; ACN, acetonitrile; Boc, tert-butoxycarbonyl; DCM, dichloromethane; DIPEA, diisopropylethylamine; DMF, N,N′-dimethylformamide; DTP, 2,2′-dithiopyridine; ESI–MS, electrospray ionization mass spectrometry; Fmoc, fluorenylmethoxycarbonyl; HATU, 1[Bis(dimethylamino)methylene]-1H-1,2,3-triazolo[4,5-b]pyridinium 3-oxid hexafluorophosphate; LC/MS, liquid chromatography/mass spectrometry; MeOH, methanol; nAChR, nicotinic acetylcholine receptor; NMR, nuclear magnetic resonance; Pbf, pentamethyl-dihydrobenzofuran-5-sulfonyl; RP-HPLC, reversed-phase high performance liquid chromatography; SPPS, solid-phase peptide synthesis; t-Bu, tert-butyl; TFA, trifluoroacetic acid; TIS, triisopropylsilane; Tris, 2-Amino-2-(hydroxymethyl)propane-1,3-diol;Trt, trityl; UV, ultra-violet.

### 4.2. Chemical Synthesis

DMF, DIPEA, ACN, TIS, TFA, piperidine and all other reagents were obtained from Sigma-Aldrich (Saint-Louis, MI, USA) or Merck (Darmstadt, Germany) and were used as supplied. Fmoc (L) amino acid derivatives and HATU were purchased from Iris Biotech (Marktredwitz, Germany). PS-2-Chlorotrityl chloride resin (100-200 mesh, 1.6 mmol/g) was purchased from Iris Biotech (Marktredwitz, Germany). The following side-chain protecting groups were used: Trt for Asn, II-IV Cys, His and Gln; tBu for Ser, Thr and Asp; and Acm for I-III Cys. Peptides were manually synthesized using the Fmoc-based solid-phase peptide synthesis technique as already described elsewhere [15]. All Fmoc amino acids and HATU were dissolved in DMF to reach 0.5 M. The first amino acid was coupled onto the resin for 6 h in a 1/1 (*v*/*v*) mix of DMF and DCM, with a 2.5-fold excess of amino acid and 5-fold excess of DIPEA followed by addition of methanol and further mixing for 15 min to cap any remaining reactive functionalities on the resin. The resin was washed with DMF, DCM, MeOH, and DMF. Fmoc deprotection was carried out with piperidine in DMF (1/2 *v*/*v*) twice for 3 min. Subsequent amino acids were coupled onto 0.1 mmol of prepared resin (determined loading value 0.73 mmol/g) twice for 10 min using an amino acid/HATU/DIPEA ratio of 5:5:10 relative to resin loading. DMF was used for resin washing between deprotection and coupling steps. After chain assembly was complete, the terminal Fmoc group was removed and the resin washed with DMF and DCM. Side-chain (except Acm) deprotection and cleavage from the resin was carried out by adding 10 mL of TFA/TIS/H_2_O (95/2.5/2.5 *v*/*v*/*v*) and stirring the mixture for 2.5 h at room temperature. Crude peptides were purified by preparative RP-HPLC and pure fractions were combined and freeze-dried. A two-step oxidation procedure was then carried out. The first disulfide bridge was formed between the free cysteine residues CysII-CysIV by dissolving the peptide at 0.2 mM in 50 mM Tris-HCl buffer adjusted to pH 8 and adding dropwise 7 equivalents of DTP at 10 mM in MeOH. When the reaction was complete, the reaction mixture was acidified to pH 3 and loaded onto preparative RP-HPLC and pure fractions were combined. The second disulfide bridge (CysI-CysIII) was formed by deprotection/oxidation of the Acm protecting group directly on the combined pure fractions of the mono-bridged intermediates by treatment with 20 equivalents of 10 mM iodine in H_2_O/TFA/ACN (78/2/20 *v*/*v*/*v*). When the reaction was complete, the reaction mixture was quenched with 20 mM ascorbic acid until total discoloration of the solution, acidified and purified by preparative RP-HPLC. The combined pure fractions were freeze-dried, and their purity was confirmed by LC/ESI–MS. CIC and Δ-CIC peptides were obtained with 3.2 and 3.1% yields (from resin loading), respectively (purity >95% estimated from LC/ESI–MS). The peptide content was estimated (from amino acid analysis) at 60% from dry weight.

### 4.3. Mass Spectrometry

#### 4.3.1. Solvents Used for LC/MS Were of HPLC Grade

Intermediate peptides were characterized using an LC/MS system consisting of an Alliance 2695 HPLC coupled to a ZQ mass spectrometer (Waters, Corp., Milford, MA, USA) fitted with an electrospray ionization source operated in the positive mode (ESI+) and a quadrupole mass analyzer. All the analyses were carried out using a Chromolith (Fontenay sous Bois, France) HighResolution RP-18e (4.6 × 25 mm, 15 nm pore, 1.15 µm particle size, flow rate 3.0 mL/min) column. A flow rate of 3 mL/min and a gradient of 0–100% B over 2.5 min for routine analyses and 0–30% B over 30 min for quality control of pure products were used. Eluent A was water/0.1% HCO2H, and eluent B consisted of acetonitrile/0.1% HCO2H. UV detection was performed at 214 nm. Electrospray mass spectra were acquired at a solvent flow rate of 200 µL/min. Nitrogen was used for both the nebulizing and drying gas. The data were obtained in a scan mode ranging from 100 to 1000 m/z or 250 to 1500 m/z in 0.7 s intervals.

Folded peptides were characterized using a Synapt G2-S high-resolution MS system (Waters, Corp., Milford, MA, USA) equipped with an ESI source and a hybrid Q-TOF mass analyzer configuration. Chromatographic separation was carried out at a flow rate of 0.4 mL/min on an Acquity H-Class ultrahigh performance liquid chromatography (UPLC) system (Waters, Corp., Milford, MA, USA) equipped with a Kinetex C18 100 Å column (100 mm × 2.1 mm, 2.6 µm particle size) from Phenomenex (France). The mobile phase consisted of water (solvent A) and ACN (solvent B) with both phases acidified by 0.1% (*v/v*) formic acid. Mass spectra were acquired in the positive ionization mode.

#### 4.3.2. Preparative RP-HPLC

Preparative RP-HPLC was run on a Gilson PLC 2250 Purification system (Villiers le Bel, France) instrument using a preparative column (Waters DeltaPak C18 Radial-Pak Cartridge, 100 Å, 40 × 100 mm, 15 μm particle size, flow rate 50.0 mL/min). Buffer A was 0.1% TFA in water, and buffer B was 0.1% TFA in acetonitrile.

### 4.4. Electrophysiology

Functional experiments were performed as described previously [15]. Briefly, plasmids encoding the respective nAChR subunits (rat α2, α3, α4, α6, β2, β4 nAChRs provided by Jim Patrick (Baylor College of Medicine, Houston, TX, USA) and subcloned in pNKS2; rat α1, β1, γ, δ in pSPOoD provided by Veit Witzemann (MPI for Medical Research, Heidelberg, Germany); human α9 and α10 in pT7TS provided by David Adams (Illawara Health and Medical Research Institute, Wollongong University, Wollongong, Australia); α6/α3 chimera generated in pNKS [15,16,21] were linearized and cRNA synthesized using the mMessageMachine kit (Invitrogen, ThermoFisher Scientific, Waltham, MA, USA). Fifty nanoliters of cRNA (0.1–0.5 µg/µL) was injected per oocyte (EcoCyte Bioscience (Dortmund, Germany) or a gift from Prof. Luis Pardo (MPI of Experimental Medicine, Göttingen, Germany) with a subunit ratio of 1:1 for α3β2, α3β4, α2β2, α2β4, α4β4, 5:1 for α4β2, 3:1 for α9β10 and 1:1:1 for α6/ α3β2β3. Oocytes were stored at 16 °C in sterile-filtered ND96 containing 5 µg/ml gentamicin.

After 1–4 days, two-electrode voltage clamp recordings were performed at −70 mV with a Turbo Tec 05X Amplifier (npi electronic, Tamm, Germany) and CellWorks software. Electrode resistances were less than 1 MΩ, and currents were filtered at 200 Hz and digitized at 400 Hz. The recording solution (ND115 for α9β10, ND96 in all other cases) with or without ACh was automatically applied via a custom-made magnetic valve system combined with a manifold mounted closely above the oocyte, thus allowing a fast (<300 ms) and reproducible solution exchange. Toxins were applied manually in the 50-µl measuring chamber and preincubated for 3 min. Agonist pulses (2s) were applied in 4-min intervals. Current responses were normalized to control responses before toxin application. GraphPad Prism (version 9.0) was used for data analysis, and a four-parameter logistic fit (Hill-fit) with plateaus constrained to 100% and 0% was used to generate dose–response curves. Oocytes from at least two frogs were used for each experiment.

### 4.5. NMR Structure Determination

NMR spectra were acquired using a Bruker 600 MHz AVANCE III NMR spectrometer (Bruker, Karlsruhe, Germany) equipped with a cryoprobe. Lyophilized synthetic peptides (~1.5–2 mg) were dissolved in 90% H_2_O/10% D_2_O, and 2D ^1^H-^1^H TOCSY, ^1^H-^1^H NOESY, ^1^H-^1^H DQF-COSY, ^1^H-^15^N HSQC, and ^1^H-^13^C HSQC spectra were recorded using standard Bruker pulse sequences with excitation sculpting for solvent suppression. All spectra were acquired at 290 K with an interscan delay of 1 s. A mixing time of 200–250 ms was used when acquiring NOESY spectra, and isotropic mixing periods of 80 ms for TOCSY spectra. Spectra were processed using Topspin v3.6.1 (Bruker, Billerica, MA, USA) and assigned manually in CcpNmr V2 [22]. An ensemble of structures was calculated using CYANA [23] including torsion-angle restraints generated by TALOS+ [24]. Final structures were visualized using MOLMOL [25] PyMol (The PyMOL Molecular Graphics System, Version 2.0 Schroödinger, LLC., New York, NY, USA) and UCSF Chimera 4.1 [26].

## Figures and Tables

**Figure 1 marinedrugs-19-00141-f001:**
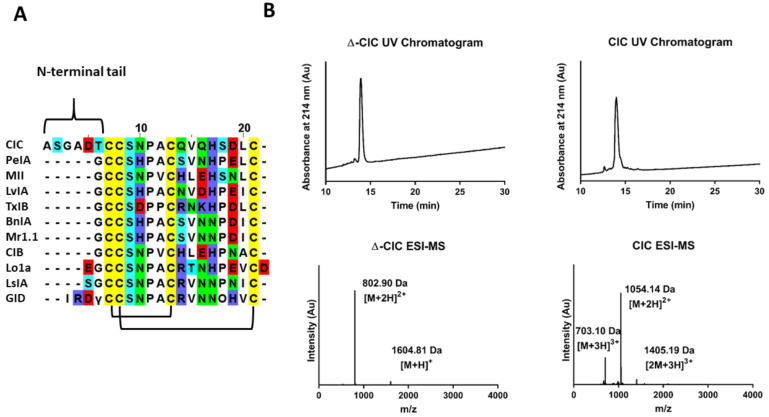
CIC alignment, UV chromatograms and ESI–MS analyses of synthetic peptides. (**A**). Alignment of CIC conotoxin with others closely related 4/7 α-conotoxins. The γ symbol indicates a carboxyglutamate residue, and O as a capital letter indicates a hydroxyproline residue. GID and CIC conotoxins are the only ones to display an N-terminal tail comprising more than two residues. The globular disulfide connectivity Cys I-III, Cys II-IV is represented. (**B**). RP-HPLC/ESI–MS analyses of synthetic ∆-CIC and CIC. ACN gradient was from 0 to 30% over 30 min. Reported mass values correspond to the average masses.

**Figure 2 marinedrugs-19-00141-f002:**
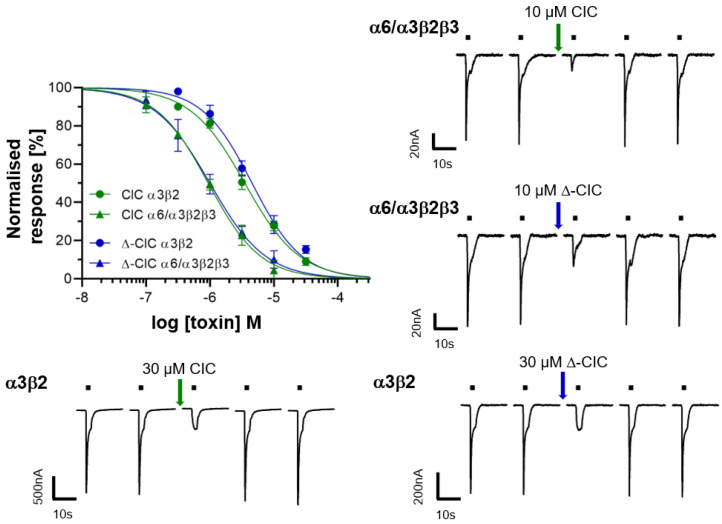
Dose–response curves and representative current traces of CIC and Δ-CIC on rat α3β2 and α6/α3β2β3 nAChR subtypes expressed in *Xenopus laevis* oocytes. 2-Electrode voltage-clamp experiments were performed at −70 mV. Responses to 2-s pulses of 100 µM ACh (indicated by a black bar) were recorded after a 3-min preincubation with the indicated peptides. Each point represents the mean of measurements from at least 3 different oocytes. Error bars represent S.D. No effects were observed on α7, α4β2, α4β4, α3β4, α2β2, α2β4 and (α1)_2_β1δγ rat subtypes as well as human α9α10 subtype at concentrations up to 10 µM.

**Figure 3 marinedrugs-19-00141-f003:**
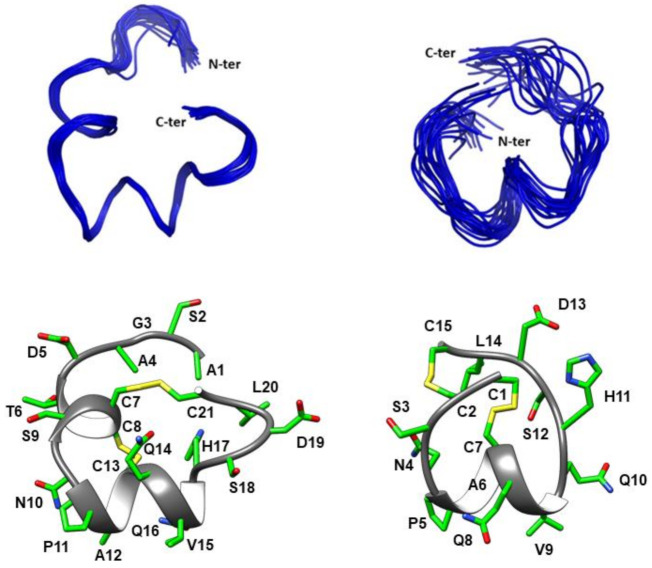
Three-dimensional structures of CIC and Δ-CIC. The backbones are shown in ribbon format and the side-chains in stick format. Top panels. Superposed backbone atoms (ribbon representation) for the 20 lowest-energy NMR structures for CIC (left) and Δ-CIC (right). Lower panels. Ribbon and stick representation of the lowest energy state for CIC (left) and Δ-CIC (right).

**Figure 4 marinedrugs-19-00141-f004:**
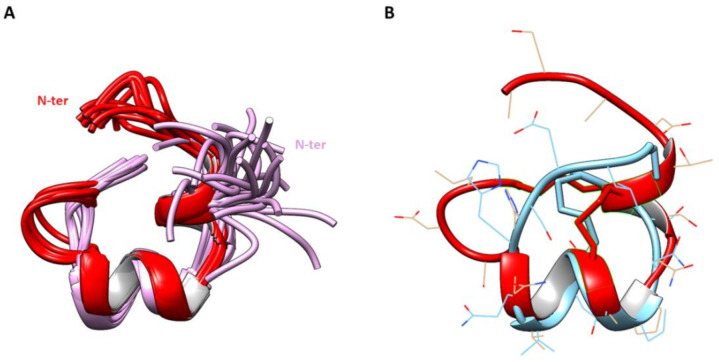
Superposition of CIC with GID and Δ-CIC. (**A**). Superimposition of the 20 lowest energy states of CIC (red) and GID (PDB: 1MTQ, mauve) over backbone atoms of residues Cys7-Cys21 (CIC numbering). Only the backbones are represented in ribbon format. (**B**). Superposition of the lowest energy state of CIC (red) and Δ-CIC (blue) over backbone atoms of residues Cys7-Cys21 (CIC numbering). The backbones are shown in ribbon format and the side-chains in stick format.

**Table 2 marinedrugs-19-00141-t002:** Structural statistics for CIC and Δ-CIC.

Experimental Restraints	CIC	Δ-CIC
Interproton distance restraints	230	88
*Intraresidue*	64	36
*Sequential*	75	45
*Medium range (i* − *j < 5)*	79	7
*Long range (i* − *j ≥5)*	12	0
Dihedral-angle restraints	30	15
**R.m.s. deviations from the mean coordinate structure (Å)**		
Backbone atoms	0.79 +/− 0.3	2.06 +/− 0.73
All heavy atoms	1.11 +/− 0.22	2.97 +/− 0.69
**Ramachandran statistics**		
% in most favoured region	82.6%	62.2%
% in additionally allowed region	17.4%	33.7%

## Data Availability

The structures presented in this paper have all been deposited in the Protein Data Bank (PDB) and Biological Magnetic Resonance Bank (BMRB) with the following codes: CIC (PDB: 7LQR) and (BMRB: 30859); Δ-CIC (PDB: 7LQS) and (BMRB: 30860). All remaining data are contained within the article.

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
