# Peer review of "Synthesis, Structural and Pharmacological Characterizations of CIC, a Novel α-Conotoxin with an Extended N-Terminal Tail"

_marinedrugs, 2021, doi:10.3390/md19030141_

Round 1
Reviewer 1 Report
This article presents the scientific results obtained by the research team in the study of the structural features and physiological activity of two new synthetic toxins of the cone snail, Conus catus.
In the course of the study, the authors not only manually synthesized by solid-phase peptide synthesis technique α-conotoxin, CIC, which is contained in the cone snail venom, but also obtained its artificial analogue with a shortened N-terminal tail, Δ-CIC. For two new conotoxins, the inhibitory ability against various subtypes of nACh receptors was assessed by electrophysiological methods, a comparative analysis of the apparent rigidity of the three-dimensional structure of toxins was performed using the high-resolution NMR spectroscopy method, three-dimensional structures were constructed, and a comparative assessment of structural features was carried out. As a result, the authors were able to prove that the change in the N-terminal part of the studied α-conotoxins does not in any way affect their ability to inhibit the α3β2- and α6/α3β2β3 nAChR subtypes.
Undoubtedly, the article is of considerable interest and could be published in Marine Drugs with minor changes.
There are minor suggestions and comments:
I would recommend the authors in the Introduction to clarify whether the new native α-conotoxin CIC was isolated from Conus catus individually and whether electrophysiological studies of native CIC for various nAChR subtypes were previously performed. If such studies have been carried out, then it is necessary to indicate the IC50 for the native CIC in Table 1 with the appropriate literature reference. If not, then you need to point it out.
Minor text formatting errors need to be addressed:
Lines 21, 62, 231. The Latin name for the cone snail should be written in Italic;
Lines 142, 147. It is necessary to correct the spelling of RMSD values;
Line 202. It is necessary to correct the spelling 1000-fold.
Table 1. It is necessary to correctly center the position of the CIC, Δ-CIC, MII and GID characters relative to the columns with numeric data.
Reviewer 2 Report
Dutertre and co-workers report the synthesis of a novel α-conotoxin, CIC, found in the predatory venom of the piscivorous species Conus catus and also, of its truncated mutant Δ-CIC, the later lacking 6 amino acids in the N-terminal tail. These compounds are small peptides with a relatively large number of disulfide bridges, which promote constriction in their structures. Peptides were synthesized through conventional solid-phase peptide techniques from commercially available Fmoc amino acids, using HATU as peptide coupling reagent, and their three-dimensional structures were determined by high-resolution NMR spectroscopy. The authors, in addition to the synthesis, also evaluated the biological activity of these peptides with respect to their ability to target neuronal and muscle-type nicotinic acetylcholine receptors, as has already been seen with other structurally similar conotoxins. The interest of these nicotinic acetylcholine receptors derives from their involvement in the processes of addiction, pain and cognition. What they found was that CIC and Δ-CIC showed similar inhibitory activities. The authors attributed to the N-terminal tail a role in stabilizing the global structure of the peptide, although it could not be ruled out that conotoxins with a larger number of amino acid units at the N-terminal side (like CIC) target receptors different from those studied in this communication.
I found this study of interest for scientists working in development of peptide therapeutics.
On the other hand, I consider the absence of NMR data and copies of NMR spectra as Supplementary Materials a weak point of this communication.
Minor corrections to be made.
- Page 4, Table 1: column headings must be aligned (centered, right…).
- Page 11, reference 26, line 441: “…Christie, M.J. (2011) a novel…” must be “…Christie, M.J. (2011) A novel…”
Reviewer 3 Report
Title: Synthesis, structural and pharmacological characterizations of 2 CIC, a novel α-conotoxin with an extended N-terminal tail
Personally, I appreciate the effort made by the authors to chemically study and synthesize a new compound present in cone snail. This conotoxin can be applied with several medical purposes.
I find the study interesting, but from my point of view it lacks an enhancement. This is observed for example in the abstract, in which the applications of this discovery are not specified.
The introduction is somewhat complex, but generally of adequate quality. There is a typographical error since a scientific name does not appear in italics. This error happens again in discussion section. Also, at the end of that paragraph there is a point left over.
In the results section, I consider it appropriate to add a figure with the reactions that take place during the synthesis of the compound to facilitate its understanding.
In the discussion, another compound is introduced that is not necessary. There are also appear several typographical errors: italics and exponents. Italics are especially striking as they are used in figure captions.
Numerous abbreviations are used throughout the text. They are explained in a paragraph, but I think it is more appropriate to do it in table form to facilitate understanding.
I am missing comparisons with other previous articles: have other authors done similar experiments? I know it is a very new theme, but another compound is mentioned that was discovered simultaneously. No further examples are given. It would be interesting to discuss this in more depth.
FINAL REMARKS
In my opinion, authors have carried out a really interesting study, with promising expectations for future research. The manuscript is clear and well written. However, I am suggesting MINOR REVISIONS. The study should be improved before publication.
